



# Ideas and perspectives: the same carbon behaves like different elements – an insight into position-specific isotope distributions

**Yuyang He[1], Xiaobin Cao[2,3], and Huiming Bao[2,3,4]**

[1]Institute of Mechanics, Chinese Academy of Sciences, No. 15 Beisihuanxi Road, Beijing 100190, PR China
[2]International Center for Isotope Effects Research, Nanjing University, Nanjing 210023, PR China
[3]School of Earth Sciences and Engineering, Nanjing University, Nanjing 210023, PR China
[4]Department of Geology and Geophysics, Louisiana State University, E235 Howe Russell Kniffen,
Baton Rouge, LA 70803, USA

**Correspondence:** Yuyang He (yhe@imech.ac.cn)

**Abstract.** It is expected that information on the source, reaction pathway, and reaction kinetics of an organic compound can be obtained from its position-specific isotope compositions or intramolecular isotope distribution (Intra-ID). To retrieve the information, we could use its predicted equilibrium Intra-ID as a reference for understanding the observed Intra-IDs. Historically, observed, apparently close-to-equilibrium carbon Intra-ID has prompted an open debate on the nature of biosystems and specifically the pervasiveness of reversible biochemical reactions. Much of the debate remains unresolved, and the discussion has not clearly distinguished between two states of equilibrium: (1) the equilibrium among the corresponding bond-breaking and bond-forming positions in reactant and product and (2) the equilibrium among all carbon positions within a compound. For an organic molecule with multiple carbon positions, equilibrium carbon Intra-ID can be attained only when a specific reaction is in equilibrium and the sources of each position are also in equilibrium with each other. An observed Intra-ID provides limited information on if the sources and pathways are both unconstrained. Here, we elaborate on this insight using examples of the observed Intra-IDs of hydroxyl-bearing minerals, $N_2O$, and acetic acid. Research effort aiming to calibrate position-specific equilibrium and kinetic isotope fractionation factors for defined processes will help to interpret observed Intra-IDs of a compound accurately and fully.

## 1 Introduction

Biosystems are dominated by a series of nonequilibrium kinetic processes. The understanding of biosystems is rooted in the study of the biochemical reaction mechanism. However, a majority of the biochemical reaction mechanisms remain elusive since they are difficult to isolate and control in laboratory experiments. Stable isotope effects can be used to examine the transition state structure and reversibility of an elementary reaction. Therefore, they can provide information on reaction mechanisms (Bigeleisen, 1949; Galimov, 2006; Bennet, 2012). However, a big organic molecule produced by an organism is the result of complex biochemical reactions that involve multiple kinetic isotope effects (KIEs) and equilibrium isotope effects (EIEs). KIE and EIE refer to the two intrinsic parameters for interpreting the observed isotope fractionations (Bao et al., 2015). According to the transition state theory (Eyring, 1935a, b), the KIE of an elementary step can be defined as the equilibrium fractionation factor between the transition state and reactant (Jones and Urbauer, 1991; Bao et al., 2015):

$$\text{KIE} = \beta_{\text{TS}}/\beta_{\text{R}}, \tag{1}$$

where the $\beta$ factor denotes the reduced partition function ratio of the transition state (TS) or reactant (R). A $\beta$ factor is the equilibrium isotope fractionation factor between an atom in a specific bond environment and its atomic form that can be predicted theoretically (Urey, 1947; Bigeleisen and Goeppert-Mayer, 1947). For a unidirectional reaction, the KIE of a reaction can also be defined as

$$\text{KIE} = {}^{h}k/{}^{l}k, \tag{2}$$

where $k$ denotes the reaction rate constant of heavy (h) or light (l) isotopes. To adapt to the convention of geochemists, we define KIE this way so that the normal KIE is less than 1.000, which is the opposite of what Bigeleisen (1949) initially defined.

EIE is the isotope fractionation among reactant and product, which is determined by the bonding environment of the target position or compound:

$$\text{EIE} = \beta_P/\beta_R, \tag{3}$$

where P denotes the product of a target reaction. It can also be defined as

$$\text{EIE} = {}^{h}K/{}^{l}K, \tag{4}$$

where $K$ denotes the equilibrium constant of a target reaction. At equilibrium, the EIE of a reaction equals the ratio of forward reaction $\text{KIE}_f$ and backward reaction $\text{KIE}_b$ ($\text{EIE} = \text{KIE}_f/\text{KIE}_b$; Bao et al., 2016).

An organic compound usually contains multiple positions of the same element, such as carbon, hydrogen, oxygen, or nitrogen. Compound-specific isotope composition refers to the bulk isotope composition of an element in an individual compound. Position-specific isotope composition refers to the isotope composition of an element at structurally distinct atomic positions within an individual compound. Information on sources, reaction pathways, and reaction kinetics of an organic molecule is pertinent to each position. The compound-specific isotope composition averages isotope compositions of all different positions of the same element in a compound, where information contained in position-specific isotope compositions could be lost (Elsner, 2010; Piasecki et al., 2018).

We named position-specific isotope compositions in a compound *intramolecular isotope distribution* or *Intra-ID* (He et al., 2018, 2020). Carbon Intra-ID in organic compounds has invoked a long-standing debate about its fundamental controls. When faced with the observed diverse Intra-IDs, earlier researchers inferred that the patterns "must be the expression of some logical order" (Schmidt, 2003), which is controlled by the EIE and KIE of biochemical reactions (e.g., Hayes, 2001, 2004; Galimov, 2009; Schmidt et al., 2015; Eiler et al., 2018; Gilbert et al., 2019). The Intra-ID was described as being in a "thermodynamic order" or "statistical isotope pattern" when each position in a molecule reaches equilibrium with each other (Galimov, 1985; Schmidt et al., 2015). Here, we name it *equilibrium Intra-ID*. The nonequilibrium state is expected to be a norm for a biochemical system since life is a dissipative system. At equilibrium, the difference in isotope composition between two positions depends on temperature only, and therefore the deviation of an observed Intra-ID from its predicted equilibrium state has been considered as an ideal reference for interpreting position-specific isotope compositions (Galimov, 1985; Hayes, 2001, 2004; He et al., 2018, 2020; Rustad, 2009; Piasecki et al., 2016).

It has been reported that different carbon fragments of chlorophyll, different carbon positions in acetoin, malonic acid, citric acid, and purine alkaloid have $^{13}\beta - \delta^{13}\text{C}$ correlation with regression coefficients in the range of 0.33–0.51 (Galimov, 1985, 2003, 2004, 2006, and references therein). Such a $^{13}\beta - \delta^{13}\text{C}$ correlation is written as $\delta^{13}\text{C} - \delta^{13}\text{C}_{\text{ave}} = \chi(\beta - \beta_{\text{ave}}) \times 10^3$, where $\chi$ is the regression coefficient. Galimov interpreted such observed intramolecular $^{13}\beta - \delta^{13}\text{C}$ correlations as "equilibrium-like" Intra-IDs produced from sets of reversible biochemical reactions at steady states which are not far from equilibrium. The $^{13}\beta - \delta^{13}\text{C}$ correlations were used as supporting evidence that the theorem of minimum entropy production can be applied in biochemical systems. However, other groups interpreted the fair-to-good correlation as fortuitous regardless of the presence or absence of complete reversibility of enzymatic reactions (Buchachenko, 2003, 2007; Schmidt, 2003; Schmidt et al., 2015). In contrast to these reported observed equilibrium-like Intra-IDs, measured position-specific $\delta^{13}\text{C}$ values are poorly correlated with their predicted $^{13}\beta$ values in organic molecules like glucose, nicotine, and tropine (Rossmann et al., 1991; Gleixner and Schmidt, 1997; Robins et al., 2016; Romek et al., 2016). Such an observed nonequilibrium Intra-ID has been termed a "non-statistical isotope pattern" (Rossmann et al., 1991; Gleixner and Schmidt, 1997; Schmidt, 2003; Robins et al., 2016; Romek et al., 2016). Buchachenko (2003, 2007) and Schmidt et al. (2004, 2015) argued that the observed $^{13}\beta - \delta^{13}\text{C}$ correlations are a random Intra-ID that only "simulates" the thermodynamic state, which cannot be used as evidence for biochemical reactions favoring an equilibrium state. The $^{13}\beta - \delta^{13}\text{C}$ correlation used an unweighted arithmetic mean isotope composition of all the components as the reference of a system. Strictly, only the mass-weighted isotope composition of all components should represent that of a system (Hayes, 2001). In addition, arbitrarily assigning a reference is not mathematically rigorous (He et al., 2018). Therefore, a $^{13}\beta - \delta^{13}\text{C}$ correlation cannot be used as supporting evidence for Galimov's hypothesis that the theorem of minimum entropy production applies in biochemical systems. Nevertheless, the invalidity of $^{13}\beta - \delta^{13}\text{C}$ correlations cannot fully quell the controversy on the nature of biosystems.

It should be noted that the debate on isotope equilibrium in biosystems among Galimov, Buchachenko, and Schmidt (and their colleagues) did not clearly distinguish between two states of equilibrium: (1) intermolecular isotope equilibrium among the corresponding bond-breaking and bond-forming positions in reactant and product in a defined *process* and (2) intramolecular isotope equilibrium among all carbon positions in a defined *molecule*. Such a difference might also

be overlooked when discussing the Intra-ID or the site preference (SP) value, i.e., the isotope composition difference among two positions. A fully reversible reaction is necessary for isotope equilibrium between corresponding active positions or functional groups. Similarly, a fully reversible intramolecular exchange mechanism must exist if different positions within a compound are to attain equilibrium. However, an overwhelming majority of biochemical reactions, especially in cases involving large organic molecules, have very few intramolecular exchange pathways. Here, we propose that the utility of parameters like the SP value in organic molecules could be limited before we obtain sufficient details on the source and pathway, as well as on the KIE and EIE of biochemical reactions. To elaborate this point, we present simple cases starting from hydroxyl-bearing minerals, in which oxygen occupies more than one position, and going on to the case of $N_2O$ in which unidirectional and fully reversible reactions can produce similar nitrogen Intra-IDs if there exists a symmetric precursor. After presenting the two inorganic cases, we move to examine measured carbon Intra-IDs from the literature of a simple organic molecule, acetic acid, in which Intra-IDs are pathway-dependent.

## 2 Intramolecular isotope distribution

### 2.1 Intracrystalline oxygen isotope difference – a rarely effective single mineral geothermometer

The same element, e.g., carbon, occupying different positions in a compound is not a unique feature of organic compounds. Some oxygen-bearing minerals have two or more position-specific oxygens, where the oxygen atoms occupy different positions in a mineral structure and have different chemical properties. Their isotope composition difference has been proposed as a potential single-mineral geothermometer. For example, it has been proposed that water temperature could be reconstructed from intracrystalline oxygen isotope difference in single-mineral copper sulfate pentahydrate ($CuSO_4 \cdot 5H_2O$; Götz et al., 1975), kaolinite ($Al_2Si_2O_5(OH)_4$), illite ($K_{0.65}Al_{2.0}(Al_{0.65}Si_{3.35}O_{10})(OH)_2$; Bechtel and Hoernes, 1990), or alunite ($KAl_3(SO_4)_2(OH)_6$; Arehart et al., 1992). By analyzing the isotope composition difference in different oxygens in the same minerals, the early researchers attempted to reconstruct the precipitation temperatures.

To be a single-mineral geothermometer, different oxygen sites must have attained equilibrium within the single mineral, which can be achieved when different positions in a compound have the same source or initially different sources are in equilibrium with each other. Take alunite precipitation from a solution as an example. Alunite has sulfate and hydroxyl oxygen positions in its structure that precipitate from sulfate and hydroxyl ions in the solution (Fig. 1). Alunite with an equilibrium Intra-ID can be produced from an equi-

librium precipitation process only if both the oxygen isotope compositions of sulfate and hydroxyl ions in the solution equilibrated with the same ambient water oxygen at the same temperature. However, the oxygen of sulfate does not readily exchange with that of water; the isotope equilibration time for $SO_4^{2-}$ and ambient water at Earth surface conditions is greater than $10^6$ to $10^7$ years (Lloyd, 1968; Turchyn and Schrag, 2004; Turchyn et al., 2010), while the oxygen in $OH^-$ can equilibrate with ambient water instantly and can readily exchange during alunite's later burial and diagenetic processes. Thus, the two oxygen positions in alunite can come from different sources at different temperatures, rendering alunite a flawed single-mineral geothermometer. The same is true for gypsum ($CaSO_4 \cdot 2H_2O$) in which sulfate oxygen is not in equilibrium with formation water and the crystallization water ($\cdot 2H_2O$) oxygen may be in equilibrium with a different type of water.

The use of a single-mineral geothermometer requires that oxygens at two different sites have attained equilibrium by exchanging with each other or with the same source oxygen (e.g., water). Unfortunately, this requirement is difficult to meet for most minerals. It is, therefore, not surprising that few successful single-mineral geothermometers exist if any at all.

### 2.2 Equilibrium-like Intra-ID produced by a kinetic process

For a compound with two different positions of the same element, a simple way to describe its observed Intra-ID is to report the difference between the two isotope compositions, i.e., the site preference (SP) value. The concept of SP originated from the study of nitrous oxide ($^\beta N^\alpha NO$), which is defined as the nitrogen isotope composition difference between the central nitrogen ($\delta^{15}N^\alpha$) and the terminal nitrogen ($\delta^{15}N^\beta$) (Yoshida and Toyoda, 2000). The predicted equilibrium SP value at room temperature in $N_2O$ is 45‰ (Yung and Miller, 1997; Wang et al., 2004; Cao and Liu, 2012). Although most observations fit the equilibrium prediction that $^{15}N$ preferentially enriches in the $^\alpha N$ position by 30‰–40‰ (Yoshida and Toyoda, 2000; Toyoda et al., 2002; Sutka et al., 2006), negative SP values have been observed nevertheless (Yamulki et al., 2001; Sutka et al., 2003).

In previous literature, the difference in SP values was explained by the difference in synthetic pathways associated with symmetrical or asymmetrical precursors (Schmidt et al., 2004; Toyoda et al., 2005; Sutka et al., 2006). If the precursor of $N_2O$ is symmetrical (e.g., –ONNO–, Fig. 2 left), the two nitrogens in the precursor are positionally equivalent; any prior isotope composition and fractionation difference would be erased by the symmetrical structures of the precursor. When producing $N_2O$ from a symmetrical precursor, the $^\beta N$ undergoes N–O bond cleavage and therefore has a primary isotope effect which is large, whereas the $^\alpha N$ has only a secondary isotope effect which is negligible (close to 1.000;

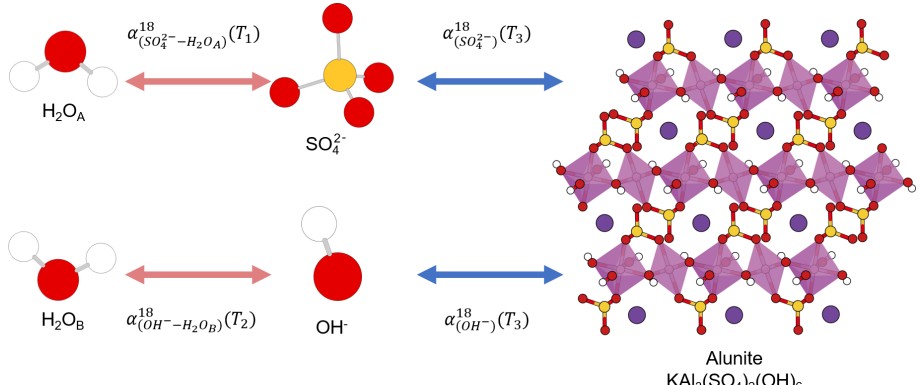

**Figure 1.** Sketch of alunite precipitation from water. The alunite could be a single-mineral geothermometer if three conditions are all fully satisfied: (1) $H_2O_A = H_2O_B$, (2) $T_1 = T_2 = T_3$, and (3) all the four reactions are fully reversible and attain equilibrium. White, red, yellow, pink, and purple spheres represent hydrogen, oxygen, sulfur, aluminum, and potassium atoms, respectively.

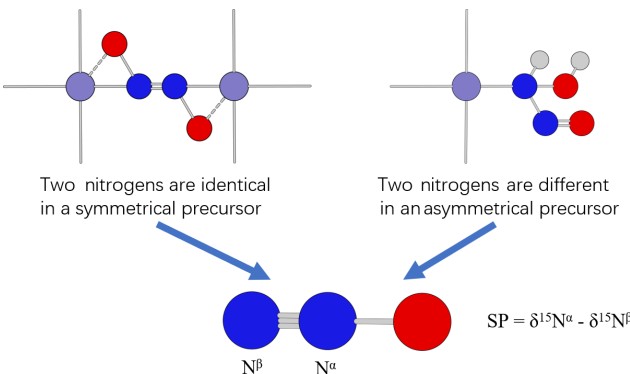

**Figure 2.** Proposed mechanisms for $N_2O$ formation from symmetrical and asymmetrical precursors (modified from Schmidt et al., 2004). Light gray, red, blue, and purple spheres represent hydrogen, oxygen, nitrogen, and iron atoms, respectively.

Bigeleisen and Wolfsberg, 1958). Therefore, $^{15}N$ depletion is expected only on the $^{\beta}N$ of $N_2O$ produced from a symmetrical precursor and is expected to have a positive SP value. If the precursor is asymmetrical (e.g., –NH(OH)NO, Fig. 2 right), the two nitrogens in the precursor are not positionally equivalent. It is assumed that the two nitrogens in the precursor were produced from different EIEs or KIEs because they went through different reaction pathways and may even have different nitrogen sources. Therefore, during the formation of $N_2O$ from an asymmetrical precursor, the difference in the position-specific $\delta^{15}N$ values of the precursors and the difference in isotope fractionation during the formation processes will be recorded in the SP value of $N_2O$. Such $N_2O$ can have either SP > 0 or SP < 0.

Nevertheless, the two previously proposed mechanisms cannot distinguish $N_2O$ with SP > 0 produced from the two mechanisms. In addition, for $N_2O$ produced from a symmetric precursor, the SP value cannot provide information on the reaction kinetics, since both fully reversible and uni-

directional reactions can produce similar observed SP values. When we state that a compound displays an equilibrium Intra-ID, the underlying assumption is that there exists a mechanism for different positions to exchange isotopes intramolecularly. However, not all observed apparent equilibrium or equilibrium-like Intra-IDs are produced by an intramolecular equilibrium process. For reactions like –ONNO– $\leftrightarrow$ $N_2O$, two types of processes could produce SP > 0. First, the $N_2O$ formation reaction is fully reversible and attains an equilibrium. When fully reversible, the two nitrogens in $N_2O$ are scrambled when it forms the symmetrical precursor through the reverse reaction. At equilibrium, the terminal nitrogen in a weaker-bond environment is expected to be depleted in heavier isotopes than the central nitrogen by 45‰ at surface temperature. Second, the $N_2O$ formation reaction is unidirectional. When unidirectional, only the N–O bond-breaking position ($^{\beta}N$) undergoes a KIE. Thus, the SP value is approximately equal to the KIE value. In this scenario, if there is a normal KIE, the terminal nitrogen is expected to be depleted in heavier isotopes than the central nitrogen by the extent of the KIE value. Such an observed Intra-ID would be similar to the predicted equilibrium Intra-ID, but it is produced by isotope depletion on the unidirectional bond-breaking process. No intramolecular exchange is involved. Therefore, even if the $N_2O$ produced by the unidirectional process has SP $\approx$ 45‰, it is not due to a close-to-equilibrium intramolecular isotope exchange. Therefore, it is necessary to distinguish between the mechanisms and reaction kinetics that can produce an observed Intra-ID.

Here we see that both fully reversible and unidirectional processes can result in a similar SP value, but the underlying mechanisms are entirely different. Furthermore, a positive SP value can also be achieved through a combination of nitrogen sources and isotope fractionations from an asymmetrical precursor. Thus, without knowing the underlying process, we cannot interpret an observed Intra-ID or SP value uniquely.

## 2.3 Position-specific isotope fractionations between reactant and product

As illustrated above, the observed Intra-ID of a compound can be used to gauge the degree of internal thermodynamic equilibrium only if we can determine the mechanisms involved in isotope fractionation. It does not mean, however, that position-specific isotope composition is useless. Based on the predicted equilibrium Intra-ID, a predicted isotope fractionation factor ($\alpha$) of corresponding positions between the reactant and product in a process can help to evaluate the thermodynamic state of a system and to decipher reaction pathways. In this section, we use a simple organic molecule, acetic acid ($CH_3COOH$), and its measured Intra-IDs from the literature CE1 as examples to illustrate how position-specific isotope fractionation occurs between reactant and product.

The relative isotope enrichment between the carboxyl and methyl carbon in acetic acid is defined as $\ln^{13}\alpha_{carb-met} = \ln(^{13}R_{carb}/^{13}R_{met})$. $^{13}R(=\ ^{13}C/^{12}C)$ denotes the carbon isotope molar abundance ratio in a position. Our calculated equilibrium Intra-ID of acetic acid has the carboxyl carbon being 47.3‰ heavier than the methyl carbon at 25 °C ($\ln^{13}\alpha_{carb-met(eq)} = 47.3$‰; He et al., 2020). The measured $\delta^{13}C_{met}$ values from literature can be lower, higher, or approximately equal to the $\delta^{13}C_{carb}$ values for acetic acids from biological, artificial, or hydrous pyrolysis samples (Table 1). The position-specific $\delta^{13}C$ values of biological, artificial, or hydrous-pyrolysis-produced acetic acid are largely overlapping in $\delta^{13}C_{met}-\delta^{13}C_{carb}$ space. For the majority of biological acetic acids, the $\delta^{13}C_{carb}$ values are several per mille higher than the $\delta^{13}C_{met}$ values (Fig. 3 top; $\ln^{13}\alpha_{carb-met} = 5.1 \pm 4.8$‰, $n = 29$), with two cases of $\sim 18$‰ higher and one case of $-2.2$‰ lower in $\delta^{13}C_{carb}$ values. It is expected that the metabolic and catabolic pathways and carbon sources are limited for most natural acetic acid. Therefore, the $\ln^{13}\alpha_{carb-met}$ value of $5.1 \pm 4.8$‰ could be characteristic but not necessarily exclusive for biologically produced acetic acid. Artificial acetic acids have a very large range of $\ln^{13}\alpha_{carb-met}$ values from $-30.2$‰ to 24.2‰ (Fig. 3 middle; $7.3 \pm 14.3$‰, $n = 24$). Biological and hydrous-pyrolysis-produced acetic acids do not have such negative $\ln^{13}\alpha_{carb-met}$ values.

Except for the abovementioned features, the production of artificial and biological acetic acid has too many unconstrained parameters. Thus, our discussion will focus on the acetic acid derived from hydrous pyrolysis of oil-prone source rocks. The acetic acids produced from the hydrous pyrolysis of oil-prone source rocks have a $\ln^{13}\alpha_{carb-met}$ value of $18.3 \pm 7.7$‰ ($n = 22$; Fig. 3 bottom). At 310–350 °C, $\ln^{13}\alpha_{carb-met}$ values of $\sim 30$‰ from Mahogany shale or Black shale with a proposed mechanism of unidirectional pyrolysis of precursor acid forms ($R-CH_2COOH \leftrightarrow R + CH_3COOH$; Fig. 4; Dias et al., 2002b). If we consider only the primary KIE between the methylene carbon in $R-*CH_2COOH$ and the methyl carbon

in acetic acid ($*CH_3COOH$), it is expected that a unidirectional process would lead to a $^{13}C$ depletion only at the methyl carbon position in acetic acid. The observed Intra-ID of the produced acetic acid should equal the $\delta^{13}C$ value difference between the precursors minus the primary KIE value. The primary KIE value is expected to be more negative than the predicted equilibrium isotope fractionation factor, which is $-14$‰ (He et al., 2020). Thus, as long as the position-specific $\delta^{13}C$ value difference between the methylene and carboxyl carbon in $R-CH_2COOH$ is greater than $-14$‰, the acetic acid produced from unidirectional pyrolysis of such precursor acid should have a carboxyl carbon with a higher $\delta^{13}C$ value than that of the methyl carbon. If the carboxyl carbon in the precursor acid has a higher $\delta^{13}C$ value than that of the methylene carbon, the pyrolysis process can easily produce acetic acid with an apparent $\ln^{13}\alpha_{carb-met}$ value close to the predicted equilibrium Intra-ID. Such apparently "equilibrium-like" Intra-ID does not involve intramolecular exchange, but it is the product of unidirectional precursor acid pyrolysis.

## 3 Implications

Life sustains itself by feeding on negative entropy. Persistent efforts are devoted to describing living systems by rigorous mathematics. Boltzmann first considered living organisms from a thermodynamic perspective, and Schrödinger later applied equilibrium thermodynamics to living systems (Popovic, 2018). Those attempts were not pursued further, since, as we all know today, a living system is an open system that is not in thermodynamic equilibrium, i.e., a dissipative system. The establishment of nonequilibrium thermodynamics by Prigogine and his coworkers has guided researchers to the theorem of minimum entropy production in biological systems (Prigogine and Wiame, 1946). The theorem of minimum entropy production in biological systems states that although a biosystem has increasing entropy, it is usually stable in a steady state. If the system is displaced from the steady state, it tends to return to its original state since the entropy is minimal at this state. Since then, efforts in applying nonequilibrium thermodynamics to living systems have been continued with mixed success (Stoward, 1962; Schneider and Kay, 1994; Hayflick, 2007; Demirel, 2010; Barbacci et al., 2015; Gerber et al., 2016). The theorem of minimum entropy production applies only to linear thermodynamic systems. Therefore, it is necessary to demonstrate that the magnitude of the reaction rate on the scale of interest in a living system is linearly dependent on the driving force responsible for the reaction system. It is reasonable to assume that a complex interacting and constantly evolving nonlinear system is constructed by a series of synergistic reactions and there should exist local linearity, a local steady state, and even local equilibrium (Galimov, 2006).

**Table 1.** Measured Intra-ID of acetic acid from the literature. TS1

| Acetic acid from biological samples | | | | Artificial acetic acid | | | | Hydrous pyrolysis of oil-prone source rocks | | | |
|---|---|---|---|---|---|---|---|---|---|---|---|
| $\delta^{13}C_{met}$ | $\delta^{13}C_{carb}$ | $\ln^{13}\alpha_{carb-met}$ | Source | $\delta^{13}C_{met}$ | $\delta^{13}C_{carb}$ | $\ln^{13}\alpha_{carb-met}$ | Source | $\delta^{13}C_{met}$ | $\delta^{13}C_{carb}$ | $\ln^{13}\alpha_{carb-met}$ | Source |
| −34.9 | −16.5 | 18.9 | Cider[a] | −15.1 | −44.4 | −30.2 | Fisher Sci., lot 725492[a] | −30.3 | −14.5 | 16.2 | Ghareb shale (Dead Sea, Israel), 350 °C[j] |
| −34.6 | −30.8 | 3.9 | Produced from ethanol by Acetobacter suboxydans[b] | −35.2 | −29.1 | 6.3 | Wako Pure Chemical Ind.[f] | −32.3 | −25.2 | 7.3 | Wilcox coal, 350 °C[j] |
| −71.1 | −69.5 | 1.7 | Produced from $CO_2$ and $HCO_3^-$ by Acetobacterium woodii[c] | −31.2 | −13.5 | 18.1 | Aldrich[f] | −37.3 | −25.8 | 11.9 | Wilcox coal, 330 °C[j] |
| −63.8 | −62.4 | 1.5 | | −29.9 | −11.4 | 18.9 | Aldrich[f] | −35 | −25.3 | 10.0 | Wilcox coal, 310 °C[j] |
| −60.5 | −62.7 | −2.2 | | −34.47 | −27.05 | 7.7 | Wako Pure Chemical Ind., lot no. TPG7942[g] | −31.8 | −24.4 | 7.6 | Wilcox coal, 280 °C[j] |
| −56.0 | −54.3 | 1.8 | | −30.96 | −13.68 | 17.7 | Aldrich, lot no. 00612PR[g] | −37.5 | −25.4 | 12.5 | Wilcox coal, 280 °C[j] |
| −52.8 | −52.3 | 0.5 | | −30.52 | −10.72 | 20.2 | Aldrich, lot no. 14002CU[g] | −37.6 | −25.2 | 12.8 | Wilcox coal, 240 °C[j] |
| −29.5 | −26.0 | 3.6 | Rice[d] | −45.01 | −26.71 | 19.0 | Wako Pure Chemical Ind., lot no. ALH4835[g] | −33.6 | −4.1 | 30.1 | Wilcox coal, 200 °C[j] |
| −28.9 | −26.5 | 2.4 | Nonglutinous brown rice[d] | −32.99 | −9.33 | 24.2 | Sigma-Aldrich, lot no. 10610MH[g] | −30.3 | −14.5 | 16.2 | Mahogany shale, 350 °C[j] |
| −29.2 | −25.6 | 3.7 | Rice[d] | −32.87 | −9.45 | 23.9 | Sigma-Aldrich, lot no. 10610MH[g] | −35.2 | −11.5 | 24.3 | Ghareb limestone (Jordan), 350 °C[j] |
| −28.7 | −25.2 | 3.6 | Rice[d] | −45.04 | −26.84 | 18.9 | Nacalai Tesque, lot no. V8M7761[g] | −36 | −13.3 | 23.3 | Ghareb limestone (Jordan), 330 °C[j] |
| −28.3 | −26.2 | 2.2 | Tomato[d] | −44.92 | −26.96 | 18.6 | Nacalai Tesque, lot no. V8M7761[g] | −36.1 | −21.1 | 15.4 | Ghareb limestone (Jordan), 310 °C[j] |
| −29.7 | −24.0 | 5.9 | Apple fruit juice[d] | −44.96 | −26.6 | 19.0 | Kishida Chemical, lot no. J98199X[g] | −34.2 | −24.1 | 10.4 | Ghareb limestone (Jordan), 280 °C[j] |
| −32.5 | −26.0 | 6.7 | Apple fruit juice[d] | −44.61 | −26.96 | 18.3 | Junsei Chemical, lot no. 815027[g] | −32.7 | −10.9 | 22.3 | Ghareb limestone (Jordan), 240 °C[j] |
| −31.5 | −24.8 | 6.9 | Wheat[d] | −44.99 | −26.59 | 19.1 | Kanto Chemical, lot no. 007W11087[g] | −32.3 | −4.7 | 28.1 | Ghareb limestone (Israel), 350 °C[j] |
| −23.8 | −24.0 | −0.3 | Lychee vinegar, grapefruit juice, lychee fruit juice[d] | −27.25 | −39.8 | −13.0 | Sigma-Aldrich[h] | −33.6 | −4.1 | 30.1 | Black shale, 350 °C[j] |
| −19.2 | −12.4 | 6.8 | Apple fruit juice, alcohol[d] | −28.7 | −34.5 | −6.0 | Distilled water[h] | −33.5 | −3.6 | 30.5 | Black shale, 330 °C[j] |
| −17.5 | −8.1 | 9.5 | Wheat, sake lees, rice, cone, alcohol[d] | −27.1 | −33.1 | −6.2 | Distilled water[h] | −33.9 | −6.4 | 28.1 | Black shale, 310 °C[j] |
| −17.7 | −11.8 | 6.0 | Sake lees, rice, alcohol[d] | −25.8 | −30.7 | −5.0 | Distilled water[h] | −34.2 | −18.4 | 16.2 | Black shale, 285 °C[j] |
| −22.0 | −10.4 | 11.8 | Sugar cane[d] | −26.4 | −31 | −4.7 | Humic acid[h] | −36.4 | −18.7 | 18.2 | New Albany shale, 330 °C[j] |

| | Acetic acid from biological samples | | | | Artificial acetic acid | | | | Hydrous pyrolysis of oil-prone source rocks | | | |
|---|---|---|---|---|---|---|---|---|---|---|---|---|
| $\delta^{13}C_{met}$ | $\delta^{13}C_{met}$ | $\delta^{13}C_{carb}$ | $\ln^{13}\alpha_{carb\text{-}met}$ | Source | $\delta^{13}C_{met}$ | $\delta^{13}C_{carb}$ | $\ln^{13}\alpha_{carb\text{-}met}$ | Source | $\delta^{13}C_{met}$ | $\delta^{13}C_{carb}$ | $\ln^{13}\alpha_{carb\text{-}met}$ | Source |
| −26.6 | −8.4 | 18.5 | Pineapple[d] | −27.6 | −31.1 | −3.6 | Raw mud[h] | −38.3 | −18.7 | 20.2 | New Albany shale, 310 °C[j] |
| −27.7 | −20.6 | 7.3 | Organic apple[e] | −28 | −30.5 | −2.6 | Distilled water[h] | −37.5 | −27.7 | 10.1 | New Albany shale, 285 °C[j] |
| −19.5 | −18.4 | 1.1 | Maize, barley[e] | −28.3 | −30.6 | −2.4 | Humic acid[h] | | | | |
| −29.6 | −26.5 | 3.2 | Wine, raspberry juice[e] | −28.5 | −29.2 | −0.7 | Raw mud[h] | | | | |
| −28.0 | −20.9 | 7.3 | Wine, cherry[e] | | | | | | | | |
| −29.3 | −25.0 | 4.4 | Wine, herbs[e] | | | | | | | | |
| −29.9 | −27.0 | 3.0 | Barley[e] | | | | | | | | |
| −28.8 | −24.5 | 4.4 | Red wine[e] | | | | | | | | |
| −29.4 | −25.4 | 4.1 | White wine[e] | | | | | | | | |

Note that $\ln^{13}\alpha_{carb\text{-}met} = \ln(^{13}R_{carb}/^{13}R_{met}) \times 1000$ is defined as the relative isotope enrichments between carboxyl and methyl carbon in acetic acid. $^{13}R = {}^{13}A/^{12}A$, and A denotes carbon isotope abundance in a position. [a] Meinschein et al. (1974). [b] Rinaldi et al. (1974). [c] Gelwicks et al. (1989). [d] Hattori et al. (2011). [e] Nimmanwudipong et al. (2015). [f] Yamada et al. (2002). [g] Yamada et al. (2014). [h] Thomas et al. (2009). [i] Dias et al. (2002a). [j] Dias et al. (2002b).

Local nonequilibrium of biochemical systems is potentially significant for the increasing complexity and ordering in the structure of life (Prigogine and Wiame, 1946; Galimov, 2006). Such a system should consist of a set of reversible reactions but not necessarily equilibrium reactions conjugated with energy supplies that are maintained in a steady state not far from equilibrium. Galimov (1985, 2004, 2006) argued that such a close-to-equilibrium steady state should be expressed as a tendency toward equilibrium inter- and intramolecular stable isotope distributions, i.e., a linear inter- and intramolecular $^{13}\beta - \delta^{13}C$ correlation with a regression coefficient smaller than but close to 1. In addition, such an equilibrium-like Intra-ID in organic molecules was proposed as a "special feature of biological systems", which could be used as a criterion to identify biologically produced extraterrestrial organic molecules (Galimov, 2003). As we have illustrated above, the observed Intra-ID in organic molecules is the product of a set of equilibrium or nonequilibrium processes as well as their source isotope compositions. An observed Intra-ID itself cannot be used as conclusive evidence for the thermodynamic state of a system. Therefore, even if a compound does have a linear intramolecular $^{13}\beta - \delta^{13}C$ correlation with a slope of 1, it does not constitute supporting evidence for the existence of an equilibrium state among biochemical reactions in organisms. To apply nonequilibrium thermodynamics to living systems, further solid evidence is needed.

A simple comparison of position-specific isotope compositions in one sample, e.g., $\ln^{13}\alpha_{carb\text{-}met}$ values of one acetic acid sample, offers little information on the reaction mechanisms and reaction kinetics of the reaction it involves. Although the position-specific atoms are the same element, without an exchange mechanism, they behave independently as different elements. It would be helpful if we could consider the position-specific atoms independently. The isotope fractionation relationship of different elements in the same compound, i.e., $(\alpha_A - 1)/(\alpha_B - 1)$, $\ln\alpha_A/\ln\alpha_B$, or $\Delta\delta_A/\Delta\delta_B$, (named the bonded isotope effect; He and Bao, 2019), is often used to characterize a reaction pathway, for instance, $\delta D$ and $\delta^{18}O$ in $H_2O$ (Dansgaard, 1964; Craig, 1961), $\delta^{15}N$ and $\delta^{18}O$ in $NO_3^-$ (Casciotti and McIlvin, 2007; Wankel et al., 2009), $\delta^{34}S$ and $\delta^{18}O$ in $SO_4^{2-}$ (Antler et al., 2013), or $\delta^{13}C$ and $\delta D$ in organic compounds (Elsner, 2010; Palau et al., 2017). The isotope composition difference in different elements is only useful if the isotope fractionation relationships are considered and their isotope compositions are normalized; e.g., $\delta(15, 18) = (\delta^{15}N - \delta^{15}N_m) - (^{15}\alpha - 1/^{18}\alpha - 1) \times (\delta^{18}O - \delta^{18}O_m)$, where $\delta^{15}N_m$ and $\delta^{18}O_m$ are the average isotope composition in a given ocean water column (Sigman et al., 2005). The normalization procedure is necessary because the source isotope compositions can affect the values of the product. Similarly, if the same element at different positions have different sources, their source isotope composition difference must also be considered. Position-specific isotope re-

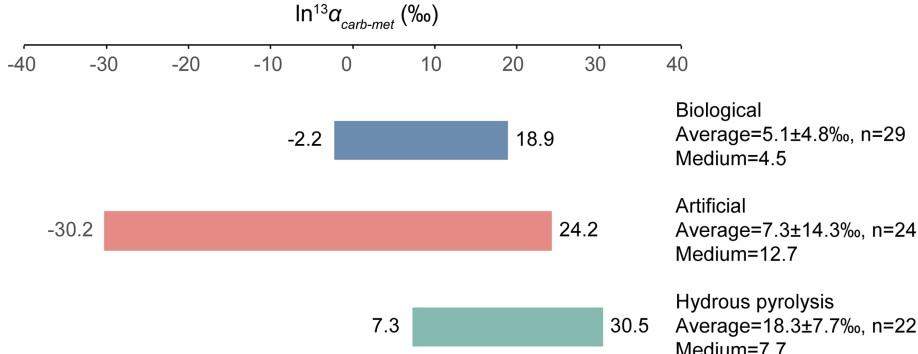

**Figure 3.** $\ln^{13}\alpha_{\text{carb-met}}$ values of biological, artificial, and hydrous-pyrolysis-produced acetic acid.

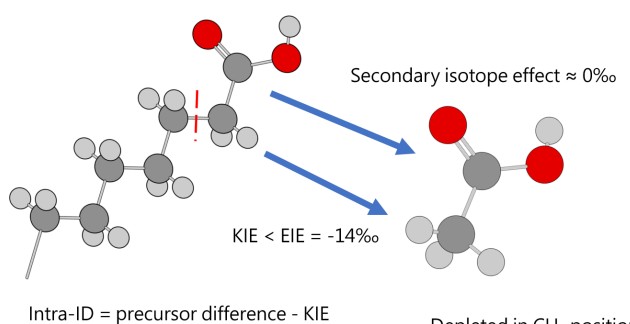

**Figure 4.** Acetic acid produced from pyrolysis of precursor acid forms has an Intra-ID that is depleted in $^{13}$C in the methyl position. Dark gray, light gray, and red spheres represent carbon, hydrogen, and oxygen atoms, respectively.

search can build upon our understanding of the bonded isotope effect.

## 4   Conclusions

An organic compound usually has an element, e.g., carbon, at different positions and therefore has an Intra-ID. The deviation of an observed Intra-ID from its equilibrium state has been used to evaluate the thermodynamic state of a system. Our analysis of oxygen-bearing minerals, $N_2O$, and acetic acids shows that both isotope sources and all reaction processes need to be in equilibrium to reach an intramolecular equilibrium state. However, such a condition is rarely satisfied. When different positions of the same element cannot exchange with each other, these different positions behave independently like different elements. Observed Intra-ID that is apparently similar to the equilibrium one can also be produced from a combination of different sources and unidirectional processes. Thus, an observed Intra-ID itself is not conclusive without adequate information on sources and reaction kinetics. Compared to position-specific isotope compositions, position-specific isotope fractionation of a defined process is more informative for identifying bond-breaking

and bond-forming positions of a large molecule, for predicting its transition state structure, for evaluating the reversibility of a biochemical process, and for determining and qualifying a process in a complex system. All in all, an understanding of a reaction process at the molecular level will always be the first step required for the later sound and wide application of stable isotope composition.

*Code and data availability.*  All data, models, and code generated or used during the study appear in the text.

*Author contributions.*  All authors conceived of the presented idea. YH collected data from the literature, drafted the manuscript, and designed the figures. XC and HB provided critical feedback and helped shape the manuscript. All authors discussed the results and contributed to the final manuscript.

*Competing interests.*  The authors declare that they have no conflict of interest.

*Financial support.*  This research has been supported by the China Postdoctoral Science Foundation (grant no. 2019M660811), the National Natural Science Foundation of China (grant no. 41490635), and the Chinese Academy of Sciences (grant no. XDB18010104).

*Review statement.*  This paper was edited by Yakov Kuzyakov and reviewed by three anonymous referees.

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

**Remarks from the language copy-editor**

CE1    We use "the literature" to denote the specific literature of a research community rather than literature in general. The use of "literatures" would be implying different sets of literature from different research fields. Please confirm it is OK to leave the text as it is or suggest alternative phrasing.

**Remarks from the typesetter**

TS1    **Editor: Please confirm insertion of Table 1. This table was missing in the submitted files.**