# Peer review of "Ideas and perspectives: Same Carbon Behave Like Different Elements- An Insight into Position-Specific Isotope Distributions"

_Biogeosciences, 2020_

## Referee Comment (RC1) · Anonymous Referee #3 · 20 Jun 2020

**1   General Comments**

In this ideas and perspectives piece, the authors discuss the limitations of interpreting position-specific isotope compositions (or intramolecular isotope distributions, "Intra-IDs") by comparison with theoretical Intra-IDs calculated for the same compound, assuming an equilibrium distribution of isotopes. In other words, if a compound is measured and displays a close-to-equilibrium distribution of isotopes, is this evidence that reversible processes dominate the synthesis of the compound? Or could the similarity to the equilibrium reference state be coincidental?

[Figure]

This work is an extension of a longstanding debate in the literature over fundamental controls on isotope distributions within molecules, especially in biological systems. Over several decades, Galimov has argued that apparent correlations between the reduced partition function ($\beta^{13}$C-factor) and isotopic compositions ($\delta^{13}$C values) of the corresponding compounds in biological systems suggest that thermodynamics, rather than kinetics or biosynthetic pathways, predominantly control the distribution of isotopes within and between biomolecules (Galimov, 1985). However, many other authors including Schmidt, Hayes, Buchachenko and colleagues cited throughout this work have argued for the importance of other, non-thermodynamic factors, which are revisited here, and that any similarities to equilibrium reference states cannot be used as evidence for biochemical reactions favoring an equilibrium state.

The central thesis of this manuscript is that Intra-IDs must be interpreted in context: sources, reaction pathways, and isotope effects (KIEs and EIEs) must be constrained in order to make sense of these Intra-ID signatures. Otherwise, any apparent similarities between position-specific isotope analyses and a predicted equilibrium state may be coincidental, actually arising from the expression of kinetic isotope effects or other features of metabolic pathways. In my opinion, this thesis is not particularly controversial and has been echoed throughout the literature, both in response to Galimov's hypothesis and elsewhere – for example, Hayes (2004) succinctly stated, "*An isotopic variation does not constitute an interpretable signal unless the mechanism controlling it is known.*" However, since a number of papers have recently highlighted the importance of predicting equilibrium Intra-IDs to provide a baseline for interpreting position-specific data (e.g., Rustad, 2009; He et al., 2018, 2020), and PSIA measurements are becoming more common (e.g., by SNIF-NMR or Fourier Transform mass spectrometry), revisiting these topics seems timely and relevant to the audience of *Biogeosciences*. The authors do a nice job of distinguishing between two concepts of equilibrium that are not sufficiently defined in some of the classic literature, the examples (oxygen-bearing minerals, $N_2O$, and acetic acid) feel appropriate for illustrating the authors' points, and the authors center a call to action to characterize more EIEs

and KIEs, sources, and pathways, for aiding interpretations moving forward.

While the authors provide sufficient literature context for their discussion, my biggest concern with the paper in its current form is that it does not clearly delineate novel insights or findings by the authors from pre-existing literature. In other words, I am not certain where the review of the historical literature ends, and the authors' analysis begins, which makes it difficult to evaluate exactly what the authors' primary contributions are here. For example, to what extent has the $N_2O$ site-preference example already been articulated in the literature? Site-preference measurements have been made for many years, and the authors cite other studies noting that precursor symmetry matters, so for readers less familiar with this application: is this simply an example compiled from existing studies to argue that more context is needed to interpret an Intra-ID, or is some component of the discussion new, like the subsequent discussion of reversibility? Some careful re-wording could clarify this and similar types of questions throughout. Similarly, it would be helpful if the authors could add at least one more sentence at the end of the oxygen-bearing minerals example synthesizing the broadly generalizable point that the authors are trying to convey.

**2  Specific Comments**

- I do not really understand the meaning behind the "same carbon different elements" portion of the title. Unless further explanation is added to the text, I think it could be easily removed. "Same carbon different positions" or something similar seems better aligned with the focus of the paper, but does not necessarily improve the existing title.

- It is not always clear throughout the text whether the term "Intra-ID" is being used to mean position-specific isotope analysis (i.e., a measurement) or a calculated distribution of isotopes (i.e., a prediction of an equilibrium state). Adding clarifying

language throughout would be helpful.

- Line 34: the definition of position-specific isotope composition is circular since it uses the words 'specific' and 'position' again. Perhaps something along the lines of "at particular atomic sites within an individual compound" or "at structurally-distinct atomic sites..." would be more clear.

- Line 38: "Intra-ID" is an abbreviation that has been introduced in prior papers, for example, He et al., 2020 GCA. It might be helpful to cite this or the earliest use of this phrase to show precedent.

- Line 39: can the authors clarify what they mean here? "Most common" in what sense? (i.e., in terms of calculations, measurements, publications, or something else)

- Line 47: "to compare to" does not add much, but could be replaced by "for interpreting position-specific isotope measurements" and I believe makes the authors' point clearer.

- Line 54: I do not think "correlate loosely" is sufficiently clear here. Perhaps, "do not correlate well with" or "are poorly correlated with"

- Line 60: the point citing He et al., 2018 seems interesting and highly relevant to the discussion in this paper. Can the authors add one more sentence summarizing the finding of that study and why the correlation approach is invalid?

- Line 97: It would be helpful to add one summary sentence at the end here to clarify what general point the authors hope the readers will take away from this section.

- Line 137: "mechanisms" might be more meaningful than "processes" here?

- In line 196, I do not understand why any references are needed. It is clear from the chemical formulae, for example, that $H_2O$ consists of H and O atoms and $NO_3^-$ consists of N and O. Why are 7 references needed in this sentence?

- I would expect to see Hayes, 2001 cited somewhere in the manuscript as another classic discussion of kinetic and metabolic controls on isotope signatures of biomolecules.

**3  Suggested minor technical corrections**

- Line 12 - "biosystem" should be plural

- Line 13 - "debates remain" should become "debate remains"

- Line 21 - "roots" → "is rooted"

- Line 24 - "to be isolated and to be controlled" → "to isolate and control"; "effect" should be plural

- Line 39 - "facing" → "faced"

- Line 42 - "termed in" → "described as being in"

- Line 53 - "contrary" → "contrast"

- Line 55 - "are also observed" is not needed

- Line 63 - "between" → "among"

- Line 71 - eliminate "in contrast to existing optimism"

- Line 95 - "can come from different sources"

- Line 179 - I believe "involving" should be "evolving" here

- Line 206 - "offer" → "offers"

- Line 218 - "at molecular level" → "at the molecular level"

---

## Referee Comment (RC2) · Anonymous Referee #2 · 23 Jun 2020

The authors propose a method to receive added value out of the knowledge of site-specific or position-specific isotopic compositions in more or less complex (organic) molecules (or inorganic minerals). They suggest to compare measured "intramolecular isotope distributions" (abbreviated as "Intra-ID") in (organic) molecules with theoretically calculated isotope distributions assuming a synthesis reaction under thermodynamic control thereby accomplishing (chemical) equilibrium. The manuscript has a sort of review character as obviously all measured intra-IDs have been taken from literature. The authors would like to test the theory that "information on the source, reaction pathway, and kinetics of an organic compound can be obtained from its position-specific isotope compositions" and end up with the familiar and already well-known conclusion

that inter- and intramolecular isotopic compositions alone are an inadequate means to reach this goal. A basic idea on the involved synthesis/breakdown reactions, flux rates and regulation points of the involved reaction pathway(s) in addition to knowledge on kinetic or equilibrium isotope effects introducing isotope fractionations and thereby sharpening the observed intra-IDs is needed. The manuscript is interesting and innovative, but needs major amendment. There might be even the need for more than this "review round". Especially, there is a need to work out a sort of a "Take home message" for the reader (in the conclusion part?). The subchapter "3 Implications" needs a complete revision. Instead of discussing equilibrium and non-equilibrium issues in terrestrial or extraterrestrial material there is need to present here which additional info is needed to interpret the Intra-IDs. Best would be here to connect e.g. N2O (site-preference) data with the schematics of a metabolism pathway producing N2O from a defined origin (and/or the acetic acid part can be elaborated in an analogues manner).

Specific comments: 1) Title: I do not understand the title. What is the meaning of "same carbon different elements" ??? –> interesting terms here might be "functional groups", "carbon molecule positions", "different bond types" ??? 2) Title: "isotope pattern" not mentioned in the text of the manuscript. What is the difference between "isotope pattern" and "isotope distribution"? Pls use only one description. 3) Line 9: Only "kinetics", no "thermodynamics" ??? kinetically controlled reactions and equilibrium, isn't that a contradiction? 4) Line 25ff: KIE give info on transition state / mechanism (rate-determining step) of a reaction, whereas EIE give info on the stiffness of the bond of the corresponding isotope in educt and product (change in bonding of the isotope in question). Your "transition-state and reversibility" is too much abridged here. Best would be to mention here in this context also the connection of KIE and EIE. EIE is equal to the ratio of the KIEs on forward and backward reaction in case the chemical (and isotopic) equilibrium has been accomplished. In kinetically controlled reactions the step between educt and the intermediary transition state (TS) is reversible and the reaction from TS towards product is irreversible. The term "equilibrium" is not helpful when talking about KIE. Please change wording correspondingly. Additionally also info on EIE would be needed here. 5) Line 30: Please use terminology of Coplen (2011, https://doi.org/10.1002/rcm.5129 ). As there are different definitions of KIE numbers used in geochemistry and other disciplines, it would be a good idea to also write the corresponding equations for KIE and EIE and clearly state which number corresponds to normal and inverse IE. Perhaps in a footnote or Appendix (as the editor recommends). A KIE of e.g. 1.01 means that the product is depleted or enriched in the heavy isotope relative to the educt? Please check with Coplen (2011) or define via own equation(s). But a definition is needed. 6) Line 36: "Pls replace "... of all different positions in a compound" by "... of all different positions of the same element in a compound". That is what you mean? 7) Line 39: There are many more paper on hydrogen isotope distribution in organic molecules. See e.g. Martin et al. (https://doi.org/10.1111/j.1365-3040.1992.tb01654.x). 13C intra ID by NMR is a relative recent approach. 8) Line 42: According to my opinion, the term "statistical" was chosen by Schmidt to explain that the distribution of the heavier isotopes in an isotopomer compounds is not a stochastic distribution but follows certain rules. In the articles by Schmidt the term "non-statistical" states that the distribution is not guided by chance, but follows a logical order. It is not stated, whether this order is under thermodynamic or kinetic control. Please adapt. In case, the text passage in italics is a direct citation, most probably Galimov or Schmidt (not both) have stated that. See also line 56. 9) Line 61: "averages": Do you mean average d-value of the whole molecule? The Intermolecular isotopic composition? 10) Line 64ff: I do not understand your differentiation between your point 1) and your point 2). Let's assume the reaction sequence ... A -> B -> C <=> D -> E -> F... (and a branching point at C and/ or D according to Hayes and Schmidt). The system should also be "regulated" on the reaction from A/B and E/F ("bottleneck" as an analogy), so that the reaction between C and D approaches or even accomplishes chemical equilibrium. The reaction between C and D should "own" an EIE (e.g. 13C EIE). Then only the carbon atoms in molecule C and molecule D can be "isotopically" equilibrated that are influenced by the primary and secondary (tertiary ??) thermodynamic isotope effects on the equilibration reaction

(Secondary isotope effects: https://goldbook.iupac.org/terms/view/S05523). It is useless (without a value, not applicable) to make a statement on the carbon atoms in C and D, that are not touched by any equilibrium isotope effects. Even secondary IE (for the heavy elements beside 2H) are normally very small. 11) Line 71: "few intramolecular exchange pathways". This statement needs either a literature citation or there is need to present own data as a proof. 12) Line 86/87: You should state here that oxygen can be bonded in different functional groups that have different chemical properties. A way out would be a position-specific analysis of the oxygen isotopic composition. 13) Line 129: Would it be possible to present a typical example for N2O produced from equilibrium or from a non-reversible reaction here? 14) Line 143: Pls define alpha with an equation (is it isotope fractionation factor ? Pls see also Coplen 2011). The factor 1000 in the alpha formula is related to the d13C formula? Meanwhile the factor 1000 is deprecated in the e.g. d13C formula. Needs to be communicated also in the text and foot note / appendix. 15) Line 151: What is the meaning of "man-made"? Produced by chemical synthesis ? 16) Line 161: "Intra-ID" should be equal to the d13C value difference between the precursor minus the primary KIE". Do you have information on the original Intra-ID of the oil from the "oil-prone source rocks"? 17) Line 162: The fact, that numbers for KIEs are higher as corresponding EIE values is commonly known. But what is a negative KIE? Please define also the equilibrium isotope fractionation factor. 18) Lines 170 to roughly 190 should be shortened. Non-essential rather distracting information is given here. The focus of the manuscript by He et al. is not to present a proof of the Galimov theory, or? 19) BTW, I do not understand the text part starting in line 196. H2O consists out of H and O, yes. Given info also true for nitrate and sulfate. There are no isotopomer water molecules. Are there isotopomer molecules of sulfate and nitrate with an Intra-ID? What idea is behind this Paragraph? It would be interesting to compare Intra-IDs of e.g. carbon and oxygen or carbon and hydrogen in organic molecules like glucose. 2H isotopomer distribution and 13C isotopomer distribution of glucose have been published already. 20) Line 44/45: It should read "Bigeleisen and Goeppert-Mayer" (with or without hyphen). Jacob Bigeleisen and Maria Goeppert

Mayer https://aip.scitation.org/doi/10.1063/1.1746492

21) Comment on the Galimov theory The above mentioned calculations for the "Equilibrium Intra IDs" are based on the framework elaborated by Galimov, who assumed that inter- and intramolecular isotope distributions in molecules of metabolic reaction networks in Nature are under thermodynamic control. The theoretically calculated b-factors (e.g. b13C for carbon) according to Galimov are compared with measured and reported d-values (e.g. d13C). The theory of Galimov on thermodynamic factors controlling the intra-IDs has been contradicted by many researchers. Additionally, to the already cited manuscripts by Buchachenko, Schmidt (and coworkers), Hayes, also Monson and Hayes (1982 Geochim Cosmochim Acat 46, 139ff), O'Leary and Yapp (1978 Biochem Biophys Res Commun 80, 155ff) and Varshavskii (1988, Biophysics 33(2), 377ff. Elsevier Pergamon Article in english) could be listed there. Dynamic reaction networks in living organism are kinetically controlled. Chemical (and isotopic) compositions of molecules at diverse levels are controlled in a steady state that allows continuous flow of mass and energy followed by a constant but adjustable flux through biochemical pathways including continuous synthesis and degradation reactions of compound molecules involved. In contrast, a system at chemical (and isotopic) equilibrium would approach a stable state and be a closed system not exchanging matter with the environment. The Gibbs free energy will then come to a minimum approaching zero. The Galimov theory is not compatible to how the biochemical pathways are explained in (plant) biochemistry text books.

––––––––––––––––––––––––––––––––

---

## Author Comment (AC1) · 2 Aug 2020

**Responses to Anonymous Referee #3**

bg-2020-120

1 General Comments

While the authors provide sufficient literature context for their discussion, my biggest concern with the paper in its current form is that it does not clearly delineate novel insights or findings by the authors from pre-existing literature. In other words, I am not certain where the review of the historical literature ends, and the authors' analysis begins, which makes it difficult to evaluate exactly what the authors' primary contributions are here. For example, to what extent has the N2O site-preference example already been articulated in the literature? Site-preference measurements have been made for many years, and the authors cite other studies noting that precursor symmetry matters, so for readers less familiar with this application: is this simply an example compiled from existing studies to argue that more context is needed to interpret an Intra-ID, or is some component of the discussion new, like the subsequent discussion of reversibility? Some careful re-wording could clarify this and similar types of questions throughout. Similarly, it would be helpful if the authors could add at least one more sentence at the end of the oxygen-bearing minerals example synthesizing the broadly generalizable point that the authors are trying to convey.

Response: Thanks for the comments. You summarized our piece accurately. Per your criticism on a lack of clear distinction of prior work and our new insights in some of the writings, we have revised the text to add phrases and sentences like "It had been proposed" "In previous literature" "The measured $\delta^{13}C_{met}$ values from literature", "Nevertheless, the two previously proposed mechanisms can only explain the observed SP<0, but cannot provide information on reaction kinetics.", "Except for the above-mentioned features, the production of artificial and biological acetic acid has too many unconstrained parameters. Thus, our discussion will focus on the acetic acid derived from hydrous pyrolysis of oil-prone source rocks."

We have added a summary at the end of the oxygen-bearing minerals example. It reads:

"The use of a single-mineral geothermometer requires that oxygens at two different sites have attained equilibrium by exchanging with each other or with the same source oxygen (e.g. water). Unfortunately, it is difficult to meet this requirement. It is, therefore, not surprising that few successful single-mineral geothermometers exist if any at all."

2 Specific Comments

I do not really understand the meaning behind the "same carbon different elements" portion of the title. Unless further explanation is added to the text, I think it could be easily removed. "Same carbon different positions" or something similar seems better aligned with the focus of the paper, but does not necessarily improve the existing title.

Response: "Same carbon different elements" here means: "The position-specific

carbons at different positions are the same element but behave like different elements." In light of the fact that the other reviewer was also puzzled by the title, we have changed our title, reluctantly, to "Carbons at Different Positions Behave Like Different Elements- An Insight into Position-Specific Isotope Distributions".

It is not always clear throughout the text whether the term "Intra-ID" is being used to mean position-specific isotope analysis (i.e., a measurement) or a calculated distribution of isotopes (i.e., a prediction of an equilibrium state). Adding clarifying language throughout would be helpful.

Response: Thanks for the suggestion. The term Intra-ID refers to the isotope distribution that is measured, predicted, or purely conceptual. To make it more clear, we have specified the measured isotope distribution as "observed/measured/apparent Intra-ID", the calculated equilibrium state as "predicted/calculated equilibrium Intra-ID".

Line 34: the definition of position-specific isotope composition is circular since it uses the words 'specific' and 'position' again. Perhaps something along the lines of "at particular atomic sites within an individual compound" or "at structurally distinct atomic sites. . ." would be more clear.

Response: Thanks for the suggestion. We have revised as suggested. It now reads:

"Position-specific isotope composition refers to the isotope composition of an element at structurally-distinct atomic positions within an individual compound."

Line 38: "Intra-ID" is an abbreviation that has been introduced in prior papers, for example, He et al., 2020 GCA. It might be helpful to cite this or the earliest use of this phrase to show precedent.

Response: Thanks for the reminder. Initially, we have defined intramolecular isotope distribution as Intra-ID, and intermolecular isotope distribution as Inter-ID in He et al., 2018 RCM. We have added the two references: He et al., 2018 RCM and He et al., 2020 GCA here.

Line 39: can the authors clarify what they mean here? "Most common" in what sense? (i.e., in terms of calculations, measurements, publications, or something else)

Response: We were trying to say that technological development in the carbon position-specific isotope analysis (PSIA) has been very active in recent years. This is in addition to the ongoing debate among Galimov, Buchachenko, and Schmidt on position-specific carbon isotope distributions. However, we realized that PSIA on hydrogen and oxygen is also developing fast. Therefore, this sentence only added confusion. We have deleted it in the new version.

Line 47: "to compare to" does not add much, but could be replaced by "for interpreting position-specific isotope measurements" and I believe makes the authors' point clearer.

Response: Thanks for the suggestion. We have revised as suggested. It now reads: "At equilibrium, the difference in isotope composition between two positions depends on temperature only and therefore such difference has been considered as a reference for interpreting position-specific isotope compositions (Galimov, 1985; Hayes, 2001, 2004; He et al., 2018, 2020; Rustad, 2009; Piasecki et al., 2016)."

Line 54: I do not think "correlate loosely" is sufficiently clear here. Perhaps, "do not correlate well with" or "are poorly correlated with"

Response: Thanks for the suggestion. We have revised it to "are poorly correlated with".

Line 60: the point citing He et al., 2018 seems interesting and highly relevant to the discussion in this paper. Can the authors add one more sentence summarizing the finding of that study and why the correlation approach is invalid?

Response: We have added a few sentences here. It now reads:

"Such a $^{13}\beta$-$\delta^{13}C$ correlation is written as $\delta^{13}C - \delta^{13}C_{ave} = \chi(\beta - \beta_{ave}) \times 10^3$, where $\chi$ is the regression coefficient. … The $^{13}\beta$-$\delta^{13}C$ correlation implicitly normalized the $^{13}\beta$ and $\delta^{13}C$ values using the averages of a given system. It revealed that the unweighted arithmetic mean isotope composition of all the components was used as the reference of a system. Strictly, only the mass-weighted isotope composition of all components should represent that of a system (Hayes, 2001). In addition, assigning an arbitrary reference is not mathematically rigorous either (He et al., 2018). Therefore, a $^{13}\beta$-$\delta^{13}C$ correlation cannot be used as supporting evidence for Galimov's hypothesis that the theorem of minimum entropy production applies in biochemical systems."

Line 97: It would be helpful to add one summary sentence at the end here to clarify what general point the authors hope the readers will take away from this section.

Response: Thanks for the suggestion. We have added a summary sentence at the end. It reads:

"The use of a single-mineral geothermometer requires that oxygens at two different sites have attained equilibrium by exchanging with each other or with the same source oxygen (e.g. water). Unfortunately, this requirement is difficult to be met for most minerals. It is, therefore, not surprising that few successful single-mineral geothermometers exist if any at all."

Line 137: "mechanisms" might be more meaningful than "processes" here?

Response: We have revised as suggested.

In line 196, I do not understand why any references are needed. It is clear from the chemical formulae, for example, that H2O consists of H and O atoms and NO− 3 consists of N and O. Why are 7 references needed in this sentence?

Response: This paragraph serves to connect to the title "Carbons at Different Positions Behave Like Different Elements- An Insight into Position-Specific Isotope Distributions". All the references are the studies of bonded isotope effects – D and $^{18}O$ in $H_2O$, $^{15}N$ and $^{18}O$ in $NO_3^-$. We have revised this paragraph to cite the references in their respective places. It now reads:

"A simple comparison of position-specific isotope compositions in one sample, e.g. $\ln^{13}\alpha_{carb-met}$ values of one acetic acid sample, offers little information on the reaction it involves. Although the position-specific atoms are the same elements, without an exchange mechanism, they behave independently as if they were different elements. The isotope fractionation relationship of different elements in the same compound, i.e. $(\alpha_A-1)/(\alpha_B-1)$, $\ln\alpha_A/\ln\alpha_B$, or $\Delta\delta_A/\Delta\delta_B$, (named bonded isotope effect, He and Bao, 2019), is useful in characterizing a reaction pathway. Some of the studied examples are $\delta D$ and $\delta^{18}O$ in $H_2O$ (Dansgaard, 1964;Craig, 1961), $\delta^{15}N$ and $\delta^{18}O$ in $NO_3^-$ (Casciotti and McIlvin, 2007; Wankel et al., 2009), $\delta^{34}S$ and $\delta^{18}O$ in $SO_4^{2-}$ (Antler et al., 2013), and $\delta^{13}C$ and $\delta D$ in organic compounds (Elsner, 2010; Palau et al., 2017). The isotope composition difference of different elements in a molecule is useful only when the isotope fractionation relationships are considered and their isotope compositions are normalized, e.g. $\Delta(15,18) = (\delta^{15}N-\delta^{15}N_m)-(^{15}\alpha-1/^{18}\alpha-1)\times(\delta^{18}O-\delta^{18}O_m)$, in which $\delta^{15}N_m$ and $\delta^{18}O_m$ are the average isotope composition in a given ocean water column (Sigman et al., 2005). The normalization procedure was necessary because the source isotope compositions can affect the values of the product. Similarly, if the same element at different positions have different sources, their source isotope composition difference must also be considered."

I would expect to see Hayes, 2001 cited somewhere in the manuscript as another classic discussion of kinetic and metabolic controls on isotope signatures of biomolecules.

Response: We have cited Hayes (2001) in multiple places
Hayes (2001) did a comprehensive review of C and H isotope fractionation in biosystems. It focuses more on biochemical processes rather than isotope effects. All the isotope fractionations discussed in Hayes (2001) are apparent ones. It has not been tested that if those values are intrinsic KIEs or EIEs. Thus, at the end of the paper, Hayes also stated: "*It often seems that isotopic fractionations provide **too much** information about **too many** processes, combining it all in a package that is unmanageably intricate.*" Hayes (2004) gave a very good example of Intra-IDs of fatty acid controlled by KIE or EIE. However, since he did not clearly define the two "equilibrium" scenarios, that example and his interpretation can be interpreted in both ways.

Although Hayes has made important contributions in the area of PSIA, we could not find a publication in which Hayes took a stand in the Galimov-Buchachenko-Schmidt debate publicly. We do not want to second guess his words, therefore, we did not

comment on Hayes' idea in this debate.

3 Suggested minor technical corrections
Line 12 - "biosystem" should be plural
Response: Corrected.

Line 13 - "debates remain" should become "debate remains"
Response: Corrected.

Line 21 - "roots" → "is rooted"
Response: Corrected.

Line 24 - "to be isolated and to be controlled" → "to isolate and control"; "effect" should be plural
Response: Corrected.

Line 39 - "facing" → "faced"
Response: Corrected.

Line 42 - "termed in" → "described as being in"
Response: Corrected.

Line 53 - "contrary" → "contrast"
Response: Corrected.

Line 55 - "are also observed" is not needed
Response: Corrected.

Line 63 - "between" → "among"
Response: Corrected.

Line 71 - eliminate "in contrast to existing optimism"
Response: Corrected.

Line 95 - "can come from different sources"
Response: Corrected.

Line 179 - I believe "involving" should be "evolving" here
Response: Corrected.

Line 206 - "offer" → "offers"
Response: Corrected.

Line 218 - "at molecular level" → "at the molecular level"
Response: Corrected.

---

## Author Comment (AC2) · 2 Aug 2020

**Responses to Anonymous Referee #2**

bg-2020-120

The authors propose a method to receive added value out of the knowledge of sitespecific or position-specific isotopic compositions in more or less complex (organic) molecules (or inorganic minerals). They suggest to compare measured "intramolecular isotope distributions" (abbreviated as "Intra-ID") in (organic) molecules with theoretically calculated isotope distributions assuming a synthesis reaction under thermodynamic control thereby accomplishing (chemical) equilibrium. The manuscript has a sort of review character as obviously all measured intra-IDs have been taken from literature. The authors would like to test the theory that "information on the source, reaction pathway, and kinetics of an organic compound can be obtained from its position-specific isotope compositions" and end up with the familiar and already well-known conclusion that inter- and intramolecular isotopic compositions alone are an inadequate means to reach this goal. A basic idea on the involved synthesis/breakdown reactions, flux rates and regulation points of the involved reaction pathway(s) in addition to knowledge on kinetic or equilibrium isotope effects introducing isotope fractionations and thereby sharpening the observed intra-IDs is needed. The manuscript is interesting and innovative, but needs major amendment. There might be even the need for more than this "review round". Especially, there is a need to work out a sort of a "Take home message" for the reader (in the conclusion part?).

Response: The central message (or take-home message) of this "Idea & Perspective" piece of ours is to convey the idea that when it comes to position-specific isotope analysis (PSIA), do not think that the carbons at different positions in a molecule behave like the same element, in fact, they behave more like different elements. In other words, most of these carbons never exchange or intermix, due to the lack of exchange mechanisms. Any attempt to treat the differently positioned carbons as they were the same element is very much like treating the O in $SO_4^{2-}$ and the O in crystallization $H_2O$ in gypsum mineral as the same O. In fact, these Cs or Os behave very much like O and S in $SO_4^{2-}$. This insight might be possessed implicitly by some in the community but was missing and should have been highlighted in the much-publicized Galimov-Buchachenko-Schmidt debate that occurred years ago. What we have read from the literature told us that it is necessary and timely to bring this insight to the open. In this contribution, we used simple inorganic and organic molecules as examples to illustrate the idea.

The subchapter "3 Implications" needs a complete revision. Instead of discussing equilibrium and non-equilibrium issues in terrestrial or extraterrestrial material there is need to present here which additional info is needed to interpret the Intra-IDs. Best would be here to connect e.g. N2O (sitepreference) data with the schematics of a metabolism pathway producing N2O from a defined origin (and/or the acetic acid part

can be elaborated in an analogues manner).

Response: The subchapter "3 Implications" serves to give the readers the background of the Galimov-Buchachenko-Schmidt debate. In Introduction, we have introduced the debate on whether there is an intramolecular $\delta^{13}C$-$^{13}\beta$ correlation or equilibrium-like isotope distribution. However, Galimov was not trying to advocate that a living system is in equilibrium or at a steady-state near equilibrium. He brought out the $\delta^{13}C$-$^{13}\beta$ correlation to support his hypothesis that the magnitude of reaction rate on the scale of interest in a living system is linearly dependent on the thermodynamic driving force responsible for the reaction system. Therefore, the theorem of minimum entropy production, i.e. Prigogine's non-equilibrium thermodynamics, can be applied in biochemical systems to explain metabolism. Although we do not think his evidence for intramolecular $\delta^{13}C$-$^{13}\beta$ correlation is sound, we do think a local thermodynamic control in an overall non-equilibrium thermodynamic system like the living system has some merits and should be mentioned here to echo what we brought out in Introduction. In addition, the last paragraph of this subchapter is to connect to the title of our manuscript that although position-specific carbons are the same element, they should be treated independently as if they were different elements in a molecule.

Specific comments:
1) Title: I do not understand the title. What is the meaning of "same carbon different elements" ??? –> interesting terms here might be "functional groups", "carbon molecule positions", "different bond types" ???

Response: "Same carbon different elements" here means: "The position-specific carbons at different positions are the same element carbon but behave like different elements." In light of the fact that the other reviewer was also puzzled by the title, we have changed our title, reluctantly, to "Carbons at Different Positions Behave Like Different Elements- An Insight into Position-Specific Isotope Distributions".

2) Title: "isotope pattern" not mentioned in the text of the manuscript. What is the difference between "isotope pattern" and "isotope distribution"? Pls use only one description.

Response: No difference, but some quoted sentences had used "pattern". We always stick to "distribution".

3) Line 9: Only "kinetics", no "thermodynamics" ??? kinetically controlled reactions and equilibrium, isn't that a contradiction?

Response: "Kinetics" here refers to reaction kinetics. We revised to use "reaction kinetics" when we referred to reaction processes in the text.
Equilibrium refers to the state of a system where the forward reaction and the backward reaction that go through the same transition state have an equal rate. Kinetic isotope

effect (KIE) and equilibrium isotope effect (EIE) is not really mutually exclusive in chemical physics. KIE by definition (in the sense of Bigeleisen & Wolfsberg, 1958 the classical paper on KIE) is the EIE between transition-state and the reactant. Currently, many isotope geochemists refer to non-equilibrium processes as "kinetically controlled", which is not a good practice; it is misleading to say the least. Partially reversible reactions at steady states can have isotope fractionations that are not at the degree of either EIE or KIE, but a combination of forward and backward KIEs. This issue has been raised by Clayton & Mayeda (2009) earlier.

4) Line 25ff: KIE give info on transition state / mechanism (rate-determining step) of a reaction, whereas EIE give info on the stiffness of the bond of the corresponding isotope in reactant and product (change in bonding of the isotope in question). Your "transition-state and reversibility" is too much abridged here. Best would be to mention here in this context also the connection of KIE and EIE. EIE is equal to the ratio of the KIEs on forward and backward reaction in case the chemical (and isotopic) equilibrium has been accomplished. In kinetically controlled reactions the step between educt and the intermediary transition state (TS) is reversible and the reaction from TS towards product is irreversible. The term "equilibrium" is not helpful when talking about KIE. Please change wording correspondingly. Additionally also info on EIE would be needed here.

Response: We agree with the reviewer on the definition of KIE and EIE, except that in the framework of transition state theory and the way KIE and EIE are calculated, there is no fundamental difference between KIE and EIE. KIE used an imaginary frequency in a transition state, which is probably the biggest difference if you do not count the difficulty of pinning down a TS structure (See also our response to comment 5 below).

A bit more information here on the issue. The forward $KIE_f$ is the EIE between transition state and reactant, and the backward $KIE_b$ is the EIE between transition state and product. "The KIE of an elementary step can be defined as the **equilibrium** fractionation factor between transition-state and reactant" is not defined by us. In Wolfsberg et al. (2009): *Isotope Effects in the Chemical, Geological, and Bio Sciences*. (DOI:10.1007/978-90-481-2265-3), Chapter 6, page 184, we quote, "Also, in TST (Chapter 4), one assumes that in a normal reaction the transition state is in chemical **equilibrium** with reactants and its concentration can be calculated from the chemical **equilibrium** constant corresponding to the reaction between the reactants and transition state." In that regard, plus our own experience, we found often that the students understood the KIE concept much more deeply when we view it in the "equilibrium" framework originally set up in Bigeleisen & Wolfsberg (1958) than in approaches adopted in most geochemistry books.

5) Line 30: Please use terminology of Coplen (2011, https://doi.org/10.1002/rcm.5129 ). As there are different definitions of KIE numbers used in geochemistry and other disciplines, it would be a good idea to also write the corresponding equations for KIE and EIE and clearly state which number corresponds to normal and inverse IE. Perhaps

in a footnote or Appendix (as the editor recommends). A KIE of e.g. 1.01 means that the product is depleted or enriched in the heavy isotope relative to the reactant? Please check with Coplen (2011) or define via own equation(s). But a definition is needed.

Response: Coplen (2011) defines KIE in the way that Bigeleisen (1949) initially defined, which is $^{light}k/^{heavy}k$. Such a definition is opposite to the equilibrium isotope effect ($^{heavy}K/^{light}K$ in both Coplen (2011) and Bigeleisen and Mayer (1947)). If we followed such a definition, it can easily cause confusion since the fractionation factor value would have different symbols. For instance, assuming the EIE of a reaction is 1.01, it means that the product is **enriched** in heavy isotope for ~10‰. Assuming the KIE of the reaction is also 1.01, by definition of Coplen (2011), it means that the product is **depleted** in heavy isotope for ~10‰. Using the concept of ε=(α-1) or lnα, the description "the EIE of the reaction is 10‰ and the KIE of the reaction is 10‰." would be confusing. Thus, in Bao (2015, GCA), he suggested follow the convention of geochemists (opposite of that of the physical organic chemists who are mostly interested in hydrogen isotopes) and define KIE as $^{heavy}k/^{light}k$ or RPFR$_{transition-state}$/RPFR$_{reactant}$, so KIE and EIE would have the same symbol (positive or negative).

Similarly, we saw suggestions of defining KIE as $^{heavy}k/^{light}k$ in a recent preprint in open discussion, Michalsk et al., 2020 (https://doi.org/10.5194/gmd-2020-114), page 6, line 34-39, in which we quote:

"*Much of the early research on KIEs were investigations of the KIE in reactions containing hydrogen isotopes and these studies usually defined a KIE = $k_L/k_H$ =$α_{L/H}$, where the k's are the rate constants for the light and heavy isotopologues. This is the inverse of the definition of α usually used in research dealing with EIE, VPIE, PHIFE and this inversion can lead to confusion. In this paper, to maintain consistency between the α values for EIE, KIE, VPIE, and PHIFE, α will be defined as heavy/light for all four effects.*"

We have revised the description and defined KIE and EIE with equations. It now reads:

"According to the transition-state theory (Eyring, 1935a, b), the KIE of an elementary step can be defined as the equilibrium fractionation factor between transition-state and reactant (Jones and Urbauer, 1991; Bao et al., 2015):

$$KIE = \beta_{TS}/\beta_R$$

where the β factor denotes the reduced partition function ratio of transition-state (TS) or reactant (R). β factor is the equilibrium isotope fractionation factor between an atom in a specific bond environment and its atomic form that can be predicted theoretically (Urey, 1947, Bigeleisen and Goeppert-Mayer, 1947). The KIE of a reaction can also be defined as:

$$KIE = {}^{h}k / {}^{l}k$$

where $k$ denotes the reaction rate constant of heavy ($h$) or light ($l$) isotopes. To adapt to

the convention of geochemists, we define KIE this way so that the normal KIE is less than 1.000, which is the opposite of what Bigeleisen (1949) initially defined.

EIE is the isotope fractionation among reactant and product, which is determined by the bonding environment of the compounds:

$$EIE = \beta_P / \beta_R$$

where $P$ denotes the product of a target reaction. It can also be defined as:

$$EIE = {}^h K / {}^l K$$

where K denotes the equilibrium constant of a target reaction. At equilibrium, the EIE of a reaction equals to the ratio of forward reaction $KIE_f$, and backward reaction $KIE_b$ (EIE = $KIE_f/KIE_b$, Bao et al., 2016)."

6) Line 36: "Pls replace ". . . of all different positions in a compound" by ". . . of all different positions of the same element in a compound". That is what you mean?

Response: Thanks for the suggestion. We have revised as suggested. It now reads:

"The compound-specific isotope composition averages isotope compositions of all different positions of the same element in a compound, where information contained in position-specific isotope compositions could be lost (Elsner, 2010; Piasecki et al., 2018)."

7) Line 39: There are many more paper on hydrogen isotope distribution in organic molecules. See e.g. Martin et al. (https://doi.org/10.1111/j.1365-3040.1992.tb01654.x). 13C intra ID by NMR is a relative recent approach.

Response: We were trying to say that technological development in the carbon position-specific isotope analysis (PSIA) has been very active in recent years. This is in addition to the ongoing debate among Galimov, Buchachenko, and Schmidt on position-specific carbon isotope distributions. However, we realized that PSIA on hydrogen and oxygen is also developing fast. Therefore, this sentence only adds confusion. We have deleted it.

8) Line 42: According to my opinion, the term "statistical" was chosen by Schmidt to explain that the distribution of the heavier isotopes in an isotopomer compounds is not a stochastic distribution but follows certain rules. In the articles by Schmidt the term "non-statistical" states that the distribution is not guided by chance, but follows a logical order. It is not stated, whether this order is under thermodynamic or kinetic control. Please adapt. In case, the text passage in italics is a direct citation, most probably Galimov or Schmidt (not both) have stated that. See also line 56.

Response: Neither Galimov nor Schmidt and their colleagues had clearly defined the terms "statistical" and "non-statistical" isotope distributions. In our understanding, these two terms refer to "equilibrium" and "non-equilibrium" isotope distributions for the following reasons:

1. In Schmidt (2003, https://doi.org/10.1007/s00114-003-0485-5), he stated that "*Nevertheless, the data elaborated and collected by Galimov (1985) show, in a number of cases, especially for the intermolecular range, more or less satisfactory correlations between $^{13}C$-contents and β factors. On the other hand, many recent investigations have proved that the **thermodynamic order** is not generally realised, especially not for intramolecular isotope distributions in natural compounds, and that unequivocally kinetic isotope effects determine the isotope abundance in many defined molecule positions. Anyway even a partial realisation of the thermodynamic order of the **nonstatistical distribution of isotopes** would demand an explanation compatible with classical enzyme kinetics.*"

   In this paragraph, he first recognized Galimov's $\delta^{13}C$-$^{13}\beta$ correlation as "*thermodynamic order*". In that context, thermodynamic order equals to equilibrium. Then, Schmidt brought out the fact that the $\delta^{13}C$-$^{13}\beta$ correlation is not common in natural compounds. In the next sentence, "*the nonstatistical distribution of isotopes*" appears following "*a partial realisation of the thermodynamic order of*". In the article, Schmidt was talking about the partial reversible biochemical process at steady-state that can produce a predictable non-equilibrium isotope distribution, which his "*nonstatistical distribution of isotopes*" must have refered to.

2. Also in Schmidt (2003), he mentioned a case of L-malate from Meinschein et al. (1984). Meinschein et al. (1984) is an abstract, and we could not find the full text of it (https://pascal-francis.inist.fr/vibad/index.php?action=getRecordDetail&idt=9067003). Nevertheless, we can see the data in Galimov (2006, https://doi.org/10.1134/S0016702906130015) in which Fig. 5.2.7 showed that the measured position-specific $\delta^{13}C$ is well correlated with predicted $^{13}\beta$. Galimov quote Meinschein: "*the $^{13}C$ contents of the specific carbon atoms in malic acid from apple and sorghum increase in accordance with their values, as predicted by Galimov.*"

[Figure]

Fig. 5.2.7. Intramolecular distribution of carbon isotopes in the malate molecule from plants C-3 (apple fruit, left line) and C-4 (sorghum, right line) (Meinschein et al., 1984).

It is also stated in Schmidt (2003): "*However, the [13]C-patterns of these acids (Fig. 3) do not at all coincide with those predicted from the precursors glucose/pyruvate (which is not contradictory to a correlation of their average $\delta^{13}C$ values), although they do show rather* **satisfactory correlations with the thermodynamic $\beta_i$ factors**, *already reported for L-malic acid by Meinschein et al. (1984).*" Therefore, we can confirm that the reported L-malic acid has an equilibrium intramolecular carbon isotope distribution. Schmidt described the equilibrium isotope distribution of L-malate as "*For "nature identical" L-malate one would expect ... a* **statistical [13]C distribution**."

3. The literature from the Schmidt group with "*Nonstatistical Carbon Isotope Distribution*" in the titles have non-equilibrium intramolecular carbon isotope distributions. For instance, "*Evidence for a* **Nonstatistical** *Carbon Isotope Distribution in Natural Glucose*" (Rossmann, Butzenlechner, and Schmidt, 1991, https://doi.org/10.1104/pp.96.2.609); "*Carbon Isotope Effects on the Fructose-1,6-bisphosphate Aldolase Reaction, Origin for* **Non-statistical** *13C Distributions in Carbohydrates*" (Gleixner and Schmidt, 1997, https://doi.org/10.1074/jbc.272.9.5382).

4. In Galimov (2006), he stated that "*Thermodynamic laws have a* **statistical** *character.*" Therefore, if an isotope distribution has "thermodynamic order", it should be described as "statistical isotope distribution"

5. Romek, et al. (2016, https://doi.org/10.1074/jbc.M116.734087) stated: "*From this, the molar fraction was calculated, which gives the extent to which the [13]C/[12]C ratios diverge from a* **statistical** *distribution.*" The equilibrium isotope distribution is the reference that has been compared to, which the measurement diverges from. Therefore, "statistical distribution" means "equilibrium distribution."

All in all, "statistical" and "non-statistical" in Schmit et al mean   "equilibrium" and "non-equilibrium" isotope distributions, respectively.

The terms "statistical" and "non-statistical" are ambiguous, therefore, in Schmidt (2015, http://dx.doi.org/10.1080/10256016.2015.1014355), he stopped using the two terms. This is also the reason we suggest to use "equilibrium" and "non-equilibrium" Intra-ID to describe isotope distributions.

9) Line 61: "averages": Do you mean average d-value of the whole molecule? The Intermolecular isotopic composition?
Response: Yes and No.

Galimov's equation is:

$$\delta^{13}C - \delta^{13}C_{ave} = \kappa(^{13}\beta - {}^{13}\beta_{ave}) \times 10^3 \quad \text{(Galimov, 1985, pg 100, eq 4.3)}$$

$\kappa$ (also written as $\chi$ in Galimov, 2004, 2006) is the regression coefficient The "ave" values are the unweighted arithmetic mean of all measured $\delta^{13}C$ or $^{13}\beta$ values, which we have criticized in He et al. (2018), from which we quote, "*In a system consisting of*

*multiple components, if we choose a component as a reference for mutual comparison, even if the reference is the average stable isotope composition of compounds of interest, we have effectively assigned that reference to be at equilibrium. The use of such a reference is not mathematically rigorous and can often be misleading when dealing with a complex non-equilibrium system. This is simply because we do not know a priori which compound or set of compounds represents the state of isotope equilibrium."* In addition, the average $\delta^{13}C$ value of the whole molecule needs to be weighted since some molecules have multiple carbons in the same position. This is the same for intermolecular isotope distribution (Hayes, 2001, pg 233, eq 5). Using the unweighted arithmetic mean is one additional problem of Galimov's $\delta^{13}C$-$^{13}\beta$ correlation.

We have revised this part. It reads:

"Such a $^{13}\beta$-$\delta^{13}C$ correlation is written as $\delta^{13}C - \delta^{13}C_{ave} = \chi(\beta - \beta_{ave}) \times 10^3$, where $\chi$ is the regression coefficient. … The $^{13}\beta$-$\delta^{13}C$ correlation implicitly normalized the $^{13}\beta$ and $\delta^{13}C$ values using the averages of a given system. It revealed that unweighted arithmetic mean isotope compositions of all the components was used as the reference of a system. Strictly, only the mass-weighted isotope composition of all components should represent that of a system (Hayes, 2001). In addition, assigning an arbitrary reference is not mathematically rigorous either (He et al., 2018). Therefore, a $^{13}\beta$-$\delta^{13}C$ correlation cannot be used as supporting evidence for Galimov's hypothesis that the theorem of minimum entropy production applies in biochemical systems."

10) Line 64ff: I do not understand your differentiation between your point 1) and your point 2). Let's assume the reaction sequence . . . A -> B -> C <=> D -> E -> F. . . (and a branching point at C and/ or D according to Hayes and Schmidt). The system should also be "regulated" on the reaction from A/B and E/F ("bottleneck" as an analogy), so that the reaction between C and D approaches or even accomplishes chemical equilibrium. The reaction between C and D should "own" an EIE (e.g. 13C EIE). Then only the carbon atoms in molecule C and molecule D can be "isotopically" equilibrated that are influenced by the primary and secondary (tertiary ??) thermodynamic isotope effects on the equilibration reaction (Secondary isotope effects: https://goldbook.iupac.org/terms/view/S05523). It is useless (without a value, not applicable) to make a statement on the carbon atoms in C and D, that are not touched by any equilibrium isotope effects. Even secondary IE (for the heavy elements beside 2H) are normally very small.

Response: Good questions and good analyses. It is precisely these questions that make our central point in the manuscript the more relevant and urgent. It looks like the notion and distinction of the two equilibrium scenarios are less clear and more difficult than we thought.

The two equilibrium scenarios are:

"1) intermolecular isotope equilibrium among the corresponding bond-breaking/forming positions in reactant and product in a defined **process**, and

2) intramolecular isotope equilibrium among all carbon positions in a defined **molecule**."

What we are talking about is one single reaction, not a reaction network, as the reviewer mentioned here. For instance, for a reaction AB ↔A'B', the process equilibrium is the equilibration among A and A', B and B'. The equilibrium among positions is the equilibrium among A and B, and among A' and B'. Yes, there will be secondary and tertiary isotope effects involved in the neighboring atoms, but those effects are respected to the specific reaction AB ↔A'B'. These neighboring atoms (e.g. carbons) themselves may not be in isotope equilibrium.

The equilibration the reviewer referred to belongs to the equilibrium scenario (1). However, when we talk about equilibrium Intra-ID, it refers to the equilibration among different (carbon) positions in a molecule, which is the scenario (2).

In the literature, when talking about an equilibrium process produces equilibrium Intra-ID, none of Galimov, Buchachenko, Schmidt, and Hayes clearly stated if they are talking about the equilibration among the corresponding positions in reactant and product or the equilibration among positions in the product. The clarification of this point is extremely important, which is the central message of our manuscript.

11) Line 71: "few intramolecular exchange pathways". This statement needs either a literature citation or there is need to present own data as a proof.

Response: We have cited a few papers illustrating the ubiquitous stability of characteristic organic carbon skeletons in Section (?) and paragraph ???. These stabilities are the reason why organic chemistry is quite predictable, and enantiomers extremely common.

12) Line 86/87: You should state here that oxygen can be bonded in different functional groups that have different chemical properties. A way out would be a position-specific analysis of the oxygen isotopic composition.

Response: We have revised this part. It now reads:

"The same element, e.g. carbon, occupies different positions in a compound is not a unique feature of organic compounds. Some oxygen-bearing minerals have two or more position-specific oxygens, where oxygen atoms occupy different positions in a mineral structure and have different chemical properties. For example, it had been proposed that water temperature could be reconstructed from intracrystalline oxygen isotope difference in a single mineral copper sulfate pentahydrate ($CuSO_4 \cdot 5H_2O$) (Götz et al., 1975), kaolinite ($Al_2Si_2O_5(OH)_4$), illite ($K_{0.65}Al_{2.0}(Al_{0.65}Si_{3.35}O_{10})(OH)_2$) (Bechtel and Hoernes, 1990), or alunite ($KAl_3(SO_4)_2(OH)_6$) (Arehart et al., 1992)."

13) Line 129: Would it be possible to present a typical example for N2O produced from equilibrium or from a non-reversible reaction here?

Response: No. We could not. None of the published data in literature can provide evidence of no reverse (backward) reaction or evidence of equal forward and backward reaction rates for $N_2O$.

14) Line 143: Pls define alpha with an equation (is it isotope fractionation factor ? Pls see also Coplen 2011). The factor 1000 in the alpha formula is related to the d13C formula? Meanwhile the factor 1000 is deprecated in the e.g. d13C formula. Needs to be communicated also in the text and foot note / appendix.

Response: Thanks for the suggestion. In the text, we wrote, "The relative isotope enrichment between the carboxyl and methyl carbon in acetic acid is defined as $\ln^{13}\alpha_{carb-met}= \ln(^{13}R_{carb}/^{13}R_{met}) \times 1000\text{‰}$". We have now deleted the redundant "$\times 1000\text{‰}$". "$\ln$" means the natural log of "$^{13}\alpha_{carb-met}$" and "$^{13}R_{carb}/^{13}R_{met}$". Here, $^{13}\alpha_{carb-met}=^{13}R_{carb}/^{13}R_{met}$, and "$\ln^{13}\alpha_{carb-met}$" is "the relative isotope enrichment between the carboxyl and methyl carbon in acetic acid". $\alpha$ is the isotope fractionation factor. We have labeled it at the first appearance of the "isotope fractionation factor" in the first paragraph of section 2.3. It now reads:

"Based on the predicted equilibrium Intra-ID, a predicted isotope fractionation factor ($\alpha$) of corresponding positions between reactant and product in a process can help to evaluate the thermodynamic state of a system and to decipher reaction pathways."

"The relative isotope enrichment between the carboxyl and methyl carbon in acetic acid is defined as $\ln^{13}\alpha_{carb-met}= \ln(^{13}R_{carb}/^{13}R_{met})$"

15) Line 151: What is the meaning of "man-made"? Produced by chemical synthesis ?

Response: We have revised all "man-made" to "artificial".

16) Line 161: "Intra-ID" should be equal to the d13C value difference between the precursor minus the primary KIE". Do you have information on the original Intra-ID of the oil from the "oil-prone source rocks"?

Response: We do not have the information on the original Intra-ID of the oil from the oil-prone source rocks. We are trying to explore the "equilibrium-like" Intra-ID expected in acetic acid produced from precursor acid pyrolysis. Therefore, that information is of no use to us.

17) Line 162: The fact, that numbers for KIEs are higher as corresponding EIE values is commonly known. But what is a negative KIE? Please define also the equilibrium isotope fractionation factor.

Response: Please see our response to your comment 3). By our definition, KIE = $^{heavy}k/^{light}k =0.98$, and the isotope enrichment ($\ln$KIE) would be -20‰, which is negative.

The extent of KIE is greater than EIE, but the number 0.98 is smaller than 0.99 (or -20‰ is smaller than -10‰). We know the whole expression has been difficult. But thanks to your comment, we have now changed the adjectives to match what specific numbers we are talking about.

18) Lines 170 to roughly 190 should be shortened. Non-essential rather distracting information is given here. The focus of the manuscript by He et al. is not to present a proof of the Galimov theory, or?

Response: See our general response.

19) BTW, I do not understand the text part starting in line 196. H2O consists out of H and O, yes. Given info also true for nitrate and sulfate. There are no isotopomer water molecules. Are there isotopomer molecules of sulfate and nitrate with an Intra-ID? What idea is behind this Paragraph? It would be interesting to compare Intra-IDs of e.g. carbon and oxygen or carbon and hydrogen in organic molecules like glucose. 2H isotopomer distribution and 13C isotopomer distribution of glucose have been published already.

Response: This paragraph is an effort to connect to the title "Same Carbons behave Like Different Elements- An Insight into Position-Specific Isotope Distributions". See our response to your general comment in the very beginning. We quote here "Any attempt to treat the differently positioned carbons as they were the same element is very much like treating the O in $SO_4^{2-}$ and the O in crystallization $H_2O$ in gypsum mineral as the same O. In fact, these Cs or Os behave very much like O and S in $SO_4^{2-}$. We have revised this paragraph. It now reads:

"A simple comparison of position-specific isotope compositions in one sample, e.g. $\ln^{13}\alpha_{carb\text{-}met}$ values of one acetic acid sample, offers little information on the reaction it involves. Although the position-specific atoms are the same elements, without an exchange mechanism, they behave independently as if they were different elements. The isotope fractionation relationship of different elements in the same compound, i.e. $(\alpha_A\text{-}1)/(\alpha_B\text{-}1)$, $\ln\alpha_A/\ln\alpha_B$, or $\Delta\delta_A/\Delta\delta_B$, (named bonded isotope effect, He and Bao, 2019), is useful in characterizing a reaction pathway. Some of the studied examples are $\delta D$ and $\delta^{18}O$ in $H_2O$ (Dansgaard, 1964; Craig, 1961), $\delta^{15}N$ and $\delta^{18}O$ in $NO_3^-$ (Casciotti and McIlvin, 2007; Wankel et al., 2009), $\delta^{34}S$ and $\delta^{18}O$ in $SO_4^{2-}$ (Antler et al., 2013), and $\delta^{13}C$ and $\delta D$ in organic compounds (Elsner, 2010; Palau et al., 2017). The isotope composition difference of different elements in a molecule is useful only when the isotope fractionation relationships are considered and their isotope compositions are normalized, e.g. $\Delta(15,18) = (\delta^{15}N\text{-}\delta^{15}N_m)\text{-}(^{15}\alpha\text{-}1/^{18}\alpha\text{-}1)\times(\delta^{18}O\text{-}\delta^{18}O_m)$, in which $\delta^{15}N_m$ and $\delta^{18}O_m$ are the average isotope composition in a given ocean water column (Sigman et al., 2005). The normalization procedure was necessary because the source isotope compositions can affect the values of the product. Similarly, if the same element at different positions have different sources, their source isotope composition difference must also be considered."

In addition, $^{18}O$ position-specific isotope analysis, i.e. $^{18}O$ distribution of glucose has been reported recently. Ma et al, (2018 https://doi.org/10.1021/acs.analchem.8b02022); Ma et al., (2020, https://doi.org/10.1021/acs.analchem.9b05314). It would be interesting to study position-specific D, $^{13}C$, and $^{18}O$ of glucose collectively.

20) Line 44/45: It should read "Bigeleisen and Goeppert-Mayer" (with or without hyphen). Jacob Bigeleisen and Maria Goeppert Mayer https://aip.scitation.org/doi/10.1063/1.1746492

Response: Thanks for the reminder, we have changed "Mayer" to "Goeppert-Mayer" both in text and in references.

21) Comment on the Galimov theory The above mentioned calculations for the "Equilibrium Intra IDs" are based on the framework elaborated by Galimov, who assumed that inter- and intramolecular isotope distributions in molecules of metabolic reaction networks in Nature are under thermodynamic control. The theoretically calculated bfactors (e.g. b13C for carbon) according to Galimov are compared with measured and reported d-values (e.g. d13C). The theory of Galimov on thermodynamic factors controlling the intra-IDs has been contradicted by many researchers. Additionally, to the already cited manuscripts by Buchachenko, Schmidt (and coworkers), Hayes, also Monson and Hayes (1982 Geochim Cosmochim Acat 46, 139ff), O'Leary and Yapp (1978 Biochem Biophys Res Commun 80, 155ff) and Varshavskii (1988, Biophysics 33(2), 377ff. Elsevier Pergamon Article in english) could be listed there. Dynamic reaction networks in living organism are kinetically controlled. Chemical (and isotopic) compositions of molecules at diverse levels are controlled in a steady state that allows continuous flow of mass and energy followed by a constant but adjustable flux through biochemical pathways including continuous synthesis and degradation reactions of compound molecules involved. In contrast, a system at chemical (and isotopic) equilibrium would approach a stable state and be a closed system not exchanging matter with the environment. The Gibbs free energy will then come to a minimum approaching zero. The Galimov theory is not compatible to how the biochemical pathways are explained in (plant) biochemistry text books.

Response: We totally agree with your account. We would like to mention two points: 1) As we explained above, Galimov was not trying to prove that the living system is at equilibrium, and local $\delta^{13}C$-$^{13}\beta$ correlation is just evidence for a biosystem evolving toward an apparent thermodynamic control. In his book (1985) and reviews (2004, 2006), he also recognized cases that $\delta^{13}C$ poorly correlated with $^{13}\beta$. 2) Previous counter-arguments against Galimov did not really nail it because those arguments failed to recognize a) the reference issue which was addressed in He et al (2018) and b) non-exchangeability issue among the many carbon positions in a large molecule. That is, true intramolecular equilibrium can hardly be achieved simply because there is no viable mechanism for the exchange, which is addressed in this manuscript.